



# Investigating sources of variability in closing the terrestrial water balance with remote sensing

Claire I. Michailovsky[1], Bert Coerver[1,2], Marloes Mul[1], Graham Jewitt[1,3,4]

[1]IHE Delft Institute for Water Education, Delft, Netherlands
[2]now at Food and Agriculture Organization of the United Nations, Rome, Italy
[3]Faculty of Civil Engineering and Geosciences, Delft University of Technology, Delft, Netherlands
[4]Centre for Water Resources Research, University of KwaZulu-Natal, Pietermaritzburg, South Africa

*Correspondence to*: Claire I. Michailovsky (c.michailovsky@un-ihe.org)

**Abstract.** Remote sensing (RS) data is becoming an increasingly important source of information for water resources management as it provides spatially distributed data on water availability and use. However, in order to guide appropriate use of the data, it is important to understand the impact of the uncertainties of RS data on water resources studies. Previous studies have shown that the degree of closure of the water balance from remote sensing data is highly variable across basins and that different RS products vary in their levels of accuracy depending on climatological and geographical conditions.

In this paper we analyzed the water balance derived runoff from global RS products for 591 catchments across the globe. We compared time-series of runoff estimated through a simplified water balance equation using 3 precipitation (CHIRPS, GPM and TRMM), 5 evapotranspiration (MODIS, SSEBop, GLEAM, CMRSET and SEBS) and 3 water storage change (GRACE-CSR, GRACE-JPL and GRACE-GFZ) RS datasets with monthly in situ discharge data for the period 2003-2016. Results were analyzed through the lens of 11 quantifiable catchment characteristics in order to investigate correlations between catchment

characteristics and the quality of RS based water balance estimates of runoff, and whether specific products performed better than others in certain conditions.

The median Nash Sutcliffe Efficiency (NSE) for all gauges and all product combinations was -0.03, and only 43.3% of the time-series reached positive NSE. A positive NSE could be obtained for 72.5% of stations with at least one product combination, while the overall best performing product combination was positive for 53.8% of stations. This confirms previous

findings that the best performing products cannot be globally established. When investigating the results by catchment characteristic, all combinations tended to show similar correlations between catchment characteristics and quality of estimated runoff, with the exception of combinations using MODIS ET for which the correlation was frequently reversed. The combinations with the GPM precipitation product performed generally worse than the CHIRPS and TRMM data. However, this can be attributed to the fact that the GPM data is available at higher latitudes compared to the other products, where

performance is generally poorer. When removing high latitude stations, this difference was eliminated and GPM and TRMM showed similar performance.



The results show the highest positive correlation between highly seasonal rainfall and runoff NSE. On the other hand, increasing snow cover, altitude and latitude all decreased the ability of the RS products to close the water balance. The catchment's dominant climate zone was also found to be correlated with time series performance with the tropical areas

providing the highest (median NSE=.11) and arid areas the lowest (median NSE=-0.09) NSE values. No correlation was found between catchment area and runoff NSE. The results point to the importance of further detailed studies on the uncertainties of the different data products and how these interact, as well as new approaches to using the data rather than simple water balance type approaches. Efforts to improve specific satellite products can also be better targeted using the results of this study.

## 1 Introduction

With increasing global population and pressure on the available water resources, it is increasingly important to understand the spatial and temporal distribution of water resources availability and use. Quantifying the components of the water balance is a necessary first step in sustainably managing resources in a river basin or catchment. However, the data available in many river basins is insufficient to make informed water management decisions. Global monitoring of discharge, which is one of the key variables of interest to water managers, has been in decline since the 1980s (Vorosmarty et al., 2001). In addition, even where

in situ data exists, accessibility of the data can be problematic.

This data gap is increasingly being filled by remote sensing products which provides many advantages (see e.g. Sheffield et al., 2018 for a full review). For instance, remote sensing data can give valuable insights into the spatial variability of water availability and consumption which can be difficult or impossible to obtain through in situ data collection. Utilizing the hydrological variables currently derived from remote sensing, it is now theoretically possible to close the water balance and

estimate runoff at the regional to global scale. However, due to uncertainties and errors in remote sensing data, this cannot currently be achieved at the scales and precision necessary for decision making (Sheffield et al., 2018).

Runoff estimation using remote sensing is typically done using some form of the following water balance equation (Eq.1) (see e.g. Syed et al., 2005):

$$R_o = P - ET_a - \frac{dS}{dt} \qquad (1)$$

where Ro is total runoff, P is the precipitation, ETa is the actual evapotranspiration and dS/dt is the total water storage change. Of the quantities in equation (1), all but the total runoff, which includes surface and subsurface components, can be derived from remote sensing at the global scale: remote sensing precipitation has been available for many years and is routinely used as input to hydrological models (see e.g. Stisen and Sandholt, 2010), ETa is not a direct RS measurement but many different algorithms have been developed to produce global scale ETa from RS data (Zhang et al., 2016), and total water storage change

can be monitored using measurements of the variation of the Earth's gravitational field by the Gravity Recovery and Climate Experiment (GRACE, Wahr et al., 2004). We note that given adequate auxiliary information (such as for example bathymetry or rating curves), discharge can be monitored using radar altimetry (see e.g. Kouraev et al., 2004; Michailovsky et al., 2012).





However, currently (2023) neither the radar altimetry nor the auxiliary information is available consistently at the global scale and in situ or modeled data is therefore necessary in order to assess the closure of the water balance using Eq.1.

A common approximation made when analyzing the terrestrial water budget using remote sensing over a hydrological basin or sub-catchment is to equate the total runoff with the discharge leaving the area of study. This is equivalent to the assumption that subsurface fluxes in and out of the basin are negligible. While this is likely to have an impact on studies at small spatial scales, it allows for the use of in situ discharge data to evaluate reliability of the remote sensing inputs to Eq. 1 which is then rewritten as Eq. (2):

$$R_o = P - ET_a - \frac{dS}{dt} \qquad (2)$$

For the components of the water cycle which are available through RS, various datasets are available and each product is subject to uncertainties and errors. These include the fact that most remote sensing measurements are indirect, therefore requiring interpretation and calibration, subject to interference (e.g. by cloud cover and topography) and limited in their spatial and temporal resolution relative to the phenomena measured. Each product uses its own algorithms, gap filling procedures

parameterization and validation methods to produce the variable of interest. Studies have shown that there is a large variability between the different products for a single variable (e.g. Sahoo et al., 2011).

Previous studies have analyzed the closure of the water balance with remote sensing and other global datasets from the regional to global scale. The first of such studies was performed by Syed et al. (2005) who used the land-atmosphere water balance to estimate discharge over the Amazon and Mississippi River Basins using data from the European Centre for Medium-Range

Weather Forecasts (ECMWF) and GRACE data to measure water storage change. They found that the total basin outflow was well correlated with observed streamflow in spite of phase (in the Amazon) and amplitude (in the Mississippi) discrepancies. Sheffield et al. (2009) also analyzed the water budget closure for the Mississippi and found that the RS-estimated discharge was greatly overestimated. Sahoo et al. (2011) estimated the water budget from remote sensing and in situ discharge gauges over 10 global river basins and found errors in the runoff estimates of the order of 5 to 25% of the mean annual precipitation

values. Both Sheffield et al. (2009) and Sahoo et al. (2011) concluded that the largest contributor to the lack of closure of the water balance were errors and biases in the precipitation products used.

At the global scale, one of the most comprehensive studies of the closure of the water balance from global products (including remote sensing products, products derived from gauges and models) was carried out by Lorenz et al. (2014). They compared the ability of combinations of 5 precipitation products (4 derived from gauges and 1 including RS and gauge measurements),

6 ET products (including MOD16 and GLEAM from RS) and 2 storage change solutions from GRACE (GFZ and CSR) over 96 catchments spread around the world. No single product combination was found to consistently outperform the others across catchments but catchments with high seasonality tended to show better results.

More recently, Lehmann et al. (2022) performed a similar analysis on 189 river basins covering 90% of the global land surface and analyzed combinations of 11 precipitation and 14 ET datasets and 11 runoff datasets (including data from land surface

models, gauge products and reanalysis datasets) and compared the computed storage change to GRACE data. They found that





95% of basins had a positive NSE for at least one product combination. They considered two catchment characteristics in analyzing their results and found that while no correlation between catchment area and closure of the water balance could be found, there was a correlation between climatic zone and performance for some of the datasets considered.

Other studies compared runoff computations obtained from different remote sensing input datasets to assess the best product
combinations in specific regions. For example, Moreira et al. (2019) computed runoff using eq. 2 over South America using 2 precipitation products (TRMM and MSWEP), 2 ET products (MOD16 and GLEAM) and 3 storage change solutions from GRACE (CSR, JPL and GFZ) and found that using GLEAM for ET estimation and MSWEP for precipitation produced the best results. They also reported that greater biases were found in semi-arid basins with low runoff coefficients.

Following the findings from previous studies that different catchment characteristics (e.g. climate and seasonality) and
different product combinations produced different results, this study aims to investigate both the ability of different combinations of RS products to reproduce in situ measurements of discharge, and to identify catchment characteristics that affect how well the closure of the water balance can be achieved. This is important in order to help guide water practitioners to choose between different remote sensing datasets as the use of RS becomes more widespread in water balance assessments as well as to better understand the sources of uncertainties present in the different products and identify areas of improvement.
In order to do this, 45 combinations of RS products (3 precipitation products, 5 ET products and 3 water storage change products) were used as input to the water balance equation (Eq. 2) and the discharge values computed were compared to discharge data collected from the Global Runoff Data Center (GRDC, 2019) over approximately 591 catchments (the number of catchments analyzed for each product combination varied due to coverage extent differences between products). The results were then analyzed using 11 quantifiable catchment characteristics to identify potential drivers of the goodness of fit between
computed and in situ values.

## 2 Methodology

The ability of different remote sensing product combinations to correctly close the water balance was assessed by deriving runoff time-series for each combination of products using the water-balance equation of a river-basin (see Eq. 2) and comparing these RS-derived runoff values with monthly time step discharge measurements obtained from the Global Runoff Data Centre
(GRDC) for a period of 14 years for which the RS products are consistently available.
The main drivers for the goodness of fit between calculated and observed runoff were investigated by evaluating 11 quantifiable basin characteristics.

### 2.1 Remote Sensing data

The data needed to solve the water balance for runoff are total water storage change, precipitation and actual evapotranspiration
(see Eq. 2) over the study period. These time series were acquired from a variety of global remote-sensing products: three



different precipitation products, five actual evapotranspiration products and three total water storage change products. An overview of these products is shown in Table 1 and details of the products are provided in the following sections.

Data was collected for a period of 14 years between 2003 and 2016, which are the full years for which the storage change from the Gravity Recovery and Climate Experiment (GRACE) data is available. All the products used are available within this

timeframe, except for CMRSET, which was discontinued at the end of 2012.

The products cover most of the globe (see spatial coverage in Table 1). CHIRPS and TRMM do not cover areas north of 50° N and south of 50° S, meaning that Antarctica and the northern parts of Canada and Russia are excluded. The spatial extent of SSEBOP is also limited to areas between 80° N and 60° S. Furthermore, it is important to note that SEBS has many missing pixels, mainly over the larger deserts, such as the Sahara and the Arabian Desert, as well as the Taiga in Canada and Russia.

All the products were re-sampled to a monthly time-scale and to a spatial resolution of 0.05° (specific methods are detailed in in the following sections). The analysis focused on spatial aggregates of runoff for catchments larger than 10,000 km$^2$ and the spatial resampling was therefore not expected to have a large impact on the results. For studies which focus on smaller scales or at the pixel-level, the impact of spatial resampling would need to be carefully considered. The choice of a monthly time scale was motivated by the timescales of the available remote sensing, in particular the GRACE dataset.

**2.1.1 Precipitation**

Different sensors and algorithms are used to estimate global precipitation from remote sensing. Many of the available precipitation products combine measurements from sensors aboard multiple satellites in order to be able to achieve higher temporal resolutions and some products are merged with in situ gauge data to improve accuracy (Sheffield et al., 2018). In this study, the following three products were used:

- The Tropical Rainfall Measuring Mission (TRMM) Multi-satellite Precipitation Analysis (TMPA) 3B42 product (Huffman et al., 2007).
   - The Climate Hazards group Infrared Precipitation with Stations (CHIRPS) version 2 product (Funk et al., 2015).
   - The Global Precipitation Measurement (GPM) mission Integrated Multi-satellitE Retrievals for GPM (IMERG) Final Run (Huffman et al., 2019).

The datasets had to be resampled from their native resolutions (see Table 1) to obtain monthly data at 0.05° spatial resolution:
   - The TRMM TMPA and GPM IMERG products were resampled to 0.05°using the nearest neighbor method.
   - The daily TRMM and CHIRPS daily data products were summed to obtain monthly values.

It should be noted that the products used are in large part computed from the same source satellite measurements. In particular, while the core GPM satellite was launched in February 2014, the IMERG algorithm was used to extend the time series back

to June 2000 using data from the TRMM satellite to produce a continuous long-term dataset. The TRMM satellite stopped operating in 2015 and, post 2015, the TMPA algorithm was applied to GPM data in order to continue producing data (Huffman, 2020).



**Table 1: Overview of the different remote-sensing products acquired**

| Product (version) | Availability | Spatial Resolution | Temporal Resolution | Spatial Coverage | Reference | Obtained from: |
|---|---|---|---|---|---|---|
| **Precipitation** | | | | | | |
| CHIRPS (v2) | 1981-present | 0.050° | Daily | 50° S-50° N | Funk et al. (2015) | https://data.chc.ucsb.edu/products/CHIRPS-2.0/ |
| TRMM TMPA (3b42 v7) | 1998-2020 | 0.25° | Daily | 50° S-50° N | Huffman et al. (2007) | https://disc2.gesdisc.eosdis.nasa.gov/opendap/TRMM_L3/TRMM_3B42_Daily.7/ |
| GPM 3IMERGDF (v06) | 2000*-present | 0.10° | Monthly | 90° N-90° S | Huffman et al. (2019) | https://gpm1.gesdisc.eosdis.nasa.gov/opendap/GPM_L3/GPM_3IMERGDF.06/ |
| **Evapotranspiration** | | | | | | |
| MOD16 A2 (v006) | 2001-present | 500m | 8-Daily | 90° N-90° S | Mu et al. (2011) | Google Earth Engine image collection: MODIS/006/MOD16A2 |
| SSEBOP (v4) | 2003-present | 0.010° | Dekadal | 80° N-60° S | Senay et al. (2013) | https://edcintl.cr.usgs.gov/downloads/sciweb1/shared/fews/web/global/monthly/eta/downloads/ |
| GLEAM (v3.3b) | 2003-2018 | 0.25° | Daily | 90° N-90° S | Miralles et al. (2011) | sftp://hydras.ugent.be (access instructions: https://www.gleam.eu/ - current accessible version: v3.6b) |
| CMRSET | 2003-2012 | 0.050° | Monthly | 90° N-90° S | Guerschman et al. (2009) | Shared by Dr. Guerschman |
| SEBS (5km Global Monthly ET) | 2003-2016 | 0.050° | Monthly | 90° N-90° S | Chen et al. (2013) | Shared by Dr. Chen |
| **Water storage change** | | | | | | |
| GRACE CSR (TELLUS_GRAC_L3_CSR_RL06_LND v6.0) | 2003-2017** | 1.0° | Monthly | 90° N-90° S | Landerer (2019a) | Retired product – see: https://podaac.jpl.nasa.gov/dataset/TELLUS_GRAC_L3_CSR_RL06_LND |
| GRACE GFZ (TELLUS_GRAC_L3_GFZ_RL06_LND v6.0) | 2003-2017** | 1.0° | Monthly | 90° N-90° S | Landerer (2019b) | Retired product – see: https://podaac.jpl.nasa.gov/dataset/TELLUS_GRAC_L3_GFZ_RL06_LND |
| GRACE JPL (TELLUS_GRAC_L3_JPL_RL06_LND v6.0) | 2003-2017** | 1.0° | Monthly | 90° N-90° S | Landerer (2019c) | Retired product – see: https://podaac.jpl.nasa.gov/dataset/TELLUS_GRAC_L3_JPL_RL06_LND |

*The TRMM mission ended in 2015, but the TMPA product continued to be produced using data from GPM, the GPM satellite was launched in 2015 but the IMERG product starts in 2000, using TRMM data.

**The GRACE mission produced data until July 2017, the GRACE-FO satellite started producing data from June 2018.





### 2.1.2 Evapotranspiration

Evapotranspiration (ET) obtained from RS data is not a direct measurement, and many different inputs are required for models to be able to represent the biophysical and environmental controls on ET (see e.g. Zhang et al., 2016). Five different evapotranspiration products have been used to solve the water balance for runoff in this study[1].

- The Operational Simplified Surface Energy Balance (SSEBop, Senay et al., 2013).
- CSIRO MODIS Reflectance-based Evapotranspiration (CMRSET, Guerschman et al., 2009).
- Global Land Evaporation Amsterdam Model (GLEAM, Miralles et al., 2011).
- Surface Energy Balance System (SEBS, Chen et al., 2013).
- MODIS Global Terrestrial Evapotranspiration Algorithm (MOD16, Mu et al., 2011).

These products use different methods and data sources for estimating evapotranspiration rates, more detail can be found in the publications listed for each product.

In order to obtain monthly data at 0.05° spatial resolution from the resolutions listed in Table 1 the following was done:

- The daily and dekadal fluxes from SSEBOP and GLEAM were summed to obtain monthly values.
- The 8-daily data from MOD16 were summed to monthly values (with reduced weights for images partially within a specific month). Missing data within a month was filled by setting the missing data to the monthly average of the available 8-day evapotranspiration in that month.
- MOD16, SSEBop and GLEAM were resampled to 0.05° using the nearest neighbor method.

### 2.1.3 Storage Change

Total water storage (the sum of surface and subsurface water storage) cannot be directly measured from remote sensing. However, Total Water Storage Anomalies (TWSA), i.e. the deviation in total water storage relative to the long term mean, can be obtained from the Gravity Recovery And Climate Experiment (GRACE) satellites which maps the Earth's gravity field approximately every 30 days (Biancamaria et al., 2019).

The TELLUS GRACE Level-3 Monthly LAND Water-Equivalent-Thickness Surface-Mass Anomaly Release 6.0 products from three processing centers were used in this study (Landerer and Swenson, 2012):

- the University of Texas – Center for Space Research (CSR, Landerer, 2019a)
- Geo Forschungs Zentrum (GFZ, Landerer, 2019b)
- Jet Propulsion Laboratory (JPL, Landerer, 2019c)

GRACE data is available between January 2003 and July 2017. The data is available in quasi-monthly time steps with variable windows of observation. However, most of the data is centered on the 16th of each month. The data was interpolated to the 16th

---

[1] Two other products were considered before being excluded from the study: the WaPOR dataset as it does not yet have global coverage, and ALEXI as it was not available to the authors at the time of the study.





day of every month and the central difference method was used to calculate the change in storage (see e.g. Biancamaria et al., 2019). Finally, the data was resampled to 0.05° using the nearest neighbor method.

## 2.2 In situ data: Global Runoff Data Centre

The RS-derived runoff was validated using observed runoff from the Global Runoff Data Centre (GRDC), whose dataset comprises more than 9,900 gauging stations all over the world. By filtering to identify stations with an upstream catchment larger than 10,000 km$^2$ and at least one record after January 1$^{st}$ 2003, an initial selection of 1,149 gauging stations was made. A large number of these stations are located in northern America, while the rest are spread out across the other continents (see Figure 1). Unfortunately, among the selected stations, there are very few stations located in some parts of the world, in particular Northern Africa, Central Asia and Southern Asia.

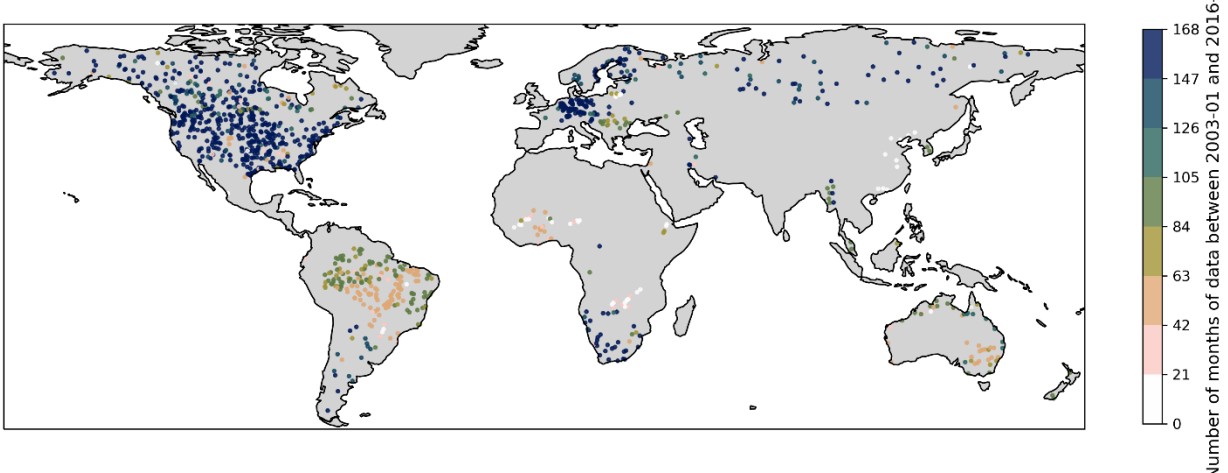

**Figure 1: Locations of the acquired GRDC stations with runoff data.**

Within the period 2003-2016, the selected stations have an average of 125 months of data, with just over half (515 stations) having more than 160 months of data out of a maximum possible of 168 months. For the first five years of this period nearly all the selected stations have data with an average of 1015 data points available each month. After 2008, the availability starts to decrease and by 2008, the average number of data points per month drops to 580. A total of 143,117 monthly runoff records were used for the analysis.

Watershed boundaries were also obtained from the GRDC (GRDC, 2011). The largest catchment covers 4,680,000 km$^2$ (the Amazon River), and most of the catchments (862) are between 10,000 and 93,600 km$^2$. The mean catchment size is 141,259 km$^2$. Altitude was known for 764 of the stations, and mean station altitude is 298.4 m.a.s.l. with a large number (161) of stations being located at altitudes below 40 m.a.s.l.

Many river basins contain multiple GRDC stations, meaning that among the 1,149 selected stations some represent nested catchments.





The monthly mean GRDC data is given in m³/s, and was converted to mm/month in order to be compared to the monthly runoff computed from remote sensing data. This was done by dividing by the catchment area.

## 2.3 Runoff time-series from remote sensing

Solving the water-balance for the different combinations of three precipitation, five actual evapotranspiration and three storage-change products, results in a total of 45 solutions. Each of these solutions consists of a series of maps of the RS-derived runoff in mm/month. For each GRDC station, the RS derived runoff time series is obtained by averaging the pixels within the corresponding catchment.

Extracting these time-series at the 1,149 locations of the selected GRDC stations from these 45 combinations gives, 51,705 time-series to analyze.

In practice the number of time series analyzed was lower due to several issues. First of all, calculated time-series that have fewer than 30 matching data points with the GRDC data were omitted. Secondly, some of the selected stations (or their catchments) are (partially) located outside of the coverage area of some of the products (see Table 1). Finally, months for which more than 20% of the pixels in a catchment were missing have been excluded (no gap-filling has been done), occasionally leading to the loss of an entire times-series (for example, as mentioned previously, SEBS has many missing pixels 225  in some parts of the world). This finally resulted in 937 locations with sufficient data.

## 2.4 Validation

The computed monthly runoff time-series have been compared with the GRDC data through the Nash–Sutcliffe model efficiency coefficient (NSE). The NSE is defined as (Nash and Sutcliffe, 1970):

$$NSE = 1 - \frac{\sum_{t=1}^{T}\left(Ro_c^t - Ro_0^t\right)^2}{\sum_{t=1}^{T}\left(Ro_0^t - \overline{Ro_o}\right)^2} \qquad (3)$$

where $\overline{Ro_0}$ is the mean of the observed runoffs, $Ro_c^t$ is the RS-derived runoff at time $t$ and $Ro_0^t$ is the observed runoff at time $t$.

## 2.5 Catchment Characteristics

We selected 11 RS derived catchment characteristics based on the findings of earlier studies to investigate correlations with quality of RS estimates of discharge. These are summarized in Table 2 and detailed below.

**Table 2: Catchment characteristics considered in this study**

| Description (continuous/discrete) | Abbreviation | Unit | Data Source |
| --- | --- | --- | --- |
| Size of the catchment (continuous) | Area | km² | GRDC (GRDC, 2019) |





| Distance of the catchment outlet to the equator (continuous) | |Latitude| | DD | GRDC (GRDC, 2019) |
|---|---|---|---|
| Altitude of the catchment outlet (continuous) | Altitude | m.a.s.l | GMTED10 |
| Total dam storage capacity in the catchment (continuous) | $S_{dam}$ | $10^6 m^3$ | GRAND (Lehner et al., 2011) |
| Seasonality: Standard deviation of the monthly precipitation in the catchment (continuous) | SDP | mm/month | GPM (Huffman et al., 2019) |
| Ratio between the mean annual runoff and the total dam storage capacity (continuous) | $\overline{Ro_{yearly}}: S_{dam}$ | − | GRAND, GRDC |
| Mean ratio between the monthly runoff and precipitation (continuous) | $R_o : P$ | − | GRDC, GPM |
| Mean of the temporal and spatial snow-cover (continuous) | $\overline{NDSI}$ | % | MOD10 (Hall et al., 2006) |
| Dominant land cover class (discrete) | LC | - | GlobCover2009 (ESA and UCLouvain, 2010) |
| Dominant climate class (discrete) | Climate | - | Köppen-Geiger Classification (Beck et al., 2018) |

*Catchment area* was chosen as a catchment parameter as it is expected that in larger catchments, the random errors may be compensated by averaging over large areas. Beyond this, the resolution of the GRACE product should also allow for better performance over larger catchments. While Biancamaria et al. (2019) found that GRACE could provide good estimates of

storage change for catchments larger than 50,000 km², most studies have considered only very large basins (>100,000 km²).

*Latitude* of the outlet of the catchment (or the distance to the equator in degrees) and *snow cover* were both chosen because precipitation products are known to have higher uncertainties at high latitudes and in the estimation of snow than in that of liquid precipitation (Tian and Peters-Lidard, 2010). Snow storage also adds a storage and therefore lag to the runoff generated in the basin which, while it should be captured by the GRACE data, can add another layer of uncertainty. ET products, in

particular those based on measurements of land surface temperature, may also face issues in computing sublimation (Xu et al., 2019).

The *altitude* of the catchment outlet is evaluated to see any difference in accuracy between river catchments with an outlet at sea level and sub-catchments with an outlet at a higher altitude. Altitude of catchment outlet is also used as a proxy for topography and precipitation products are known to have higher uncertainty over areas of rough topography (Tian and Peters-

Lidard, 2010).

*Dam storage capacity* was also considered due to the smoothing effect on the runoff. While the dam storage should be captured by the GRACE data, it has been shown that GRACE solutions do not always correctly locate the relatively punctual changes in storage due to signal leakage which could impact the results (Wang et al., 2019). Dam storage capacity relative to mean





annual runoff was also considered both as a measure of the level of modification of the basin, and as normalization for total

dam storage capacity.

The *seasonality of rainfall* varies greatly around the world. Some regions have a clear dry and wet season, while others receive rainfall throughout the entire year. In order to make a distinction between these different rainfall patterns, the standard deviation of the monthly rainfall was chosen as a parameter. A catchment with a clear wet and dry season will have a higher standard deviation than a catchment with precipitation throughout the year.

Finally, the *ratio between runoff and precipitation* is considered. Catchments with a low runoff to precipitation ratio will typically have a high evapotranspiration rate relative to precipitation, while a higher ratio indicates a low evapotranspiration rate. Catchments with ratios above 1 indicate discharge originating from either storage depletion in the basin, or inter-basin transfers.

Besides the above characteristics which can be described by continuous variables, the following two discrete characteristics

were considered:

The *dominant climate class* according to the Köppen-Geiger climate classification was computed for each catchment based on data from Beck et al. (2018). This was considered as previous water balance closure studies have shown variable performance under different climate conditions (e.g. Lorenz et al., 2014),

The final catchment characteristic considered was *dominant land cover class* in the catchment (computed from GlobCover2009

(ESA and UCLouvain, 2010)). This was considered due to the variable performance of ET products in over different land cover types (e.g. Senay et al., 2013).

For each of the continuous catchment characteristics, the Pearson coefficient *(r)* was computed to assess the correlation between the characteristic and the NSE of the discharge time series. The significance of the correlations ($p < 0.05$) was tested using a two-sided student t-test. *r* is defined as:

$$r = \frac{\sum_{i-1}^{n}\left((x_i - \bar{x})(y_i - \bar{y})\right)}{\sqrt{\sum_{i=1}^{n}(x_i - \bar{x})^2} \times \sqrt{\sum_{i=1}^{n}(y_i - \bar{y})^2}} \tag{4}$$

Where $x_i$ is the NSE for catchment $i$, $y_i$ is the catchment characteristic value for catchment $i$, and $\bar{x}$ and $\bar{y}$ are the mean NSE and catchment characteristic value.

For the two non-continuous characteristics (LULC and Climate class), the influence of the characteristic on the performance was analyzed by comparing the NSE values obtained per class.





## 3 Results and Discussion

### 3.1 Results per GRDC station

NSE values were computed for the 45 possible product combinations, for all GRDC stations possible for each combination. Figure 2 shows the median NSE value for all possible product combinations at each of the 937 GRDC stations for which at least one NSE value could be computed.

On average, for all combinations of products at all available GRDC stations, 43.4% of the generated discharge time series achieve a positive median NSE value, with only 3.2% of the discharge stations obtained a median NSE > 0.5. A positive NSE indicates a model performs better than the long-term mean of the observed time series as a predictor. Hydrological models are often considered to be of good quality when reaching NSE values of > 0.5, although many studies use different thresholds (Moriasi et al., 2007).

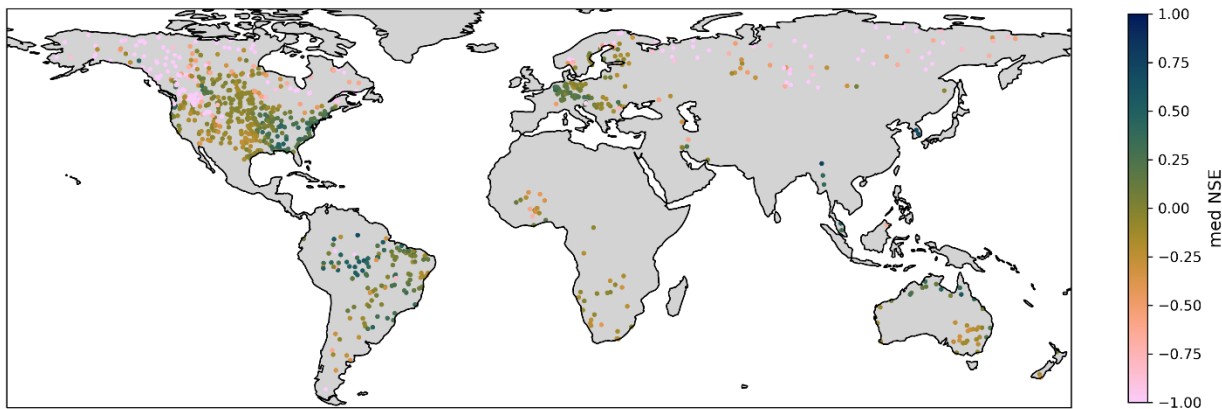

**Figure 2: Median NSE for different product combinations at each GRDC station**

When considering the maximum NSE reached at each station, it was determined that a positive NSE was reached for at least one product combination for 72.5% of the stations, and an NSE of more than 0.5 was reached for only 7% of the stations. The geographical distribution of maximum NSE values is shown in Figure 3.

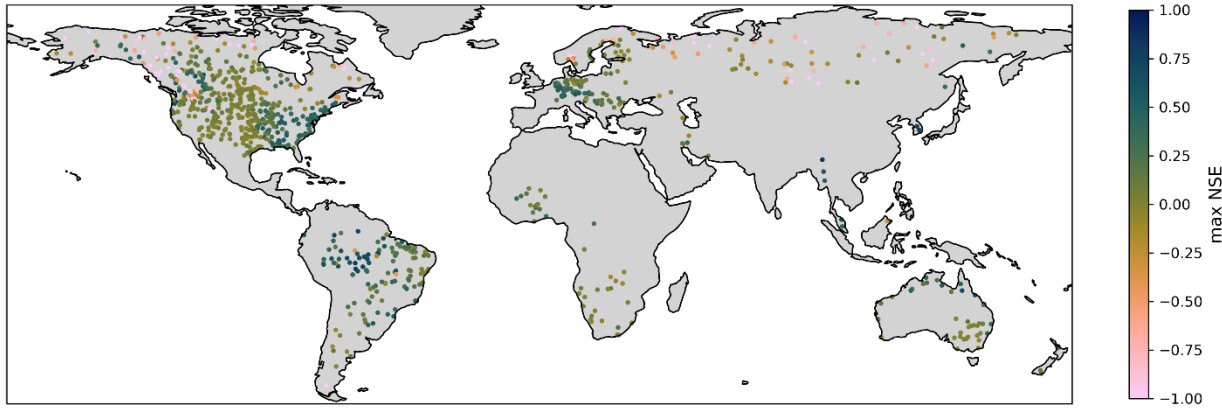





**Figure 3: Max NSE achieved at each GRDC station**

In the studies performed by Lorenz et al. (2014), positive NSE values were reached in 29 of the 96 (30%) basin considered while in the study by Lehmann et al. (2022) this was achieved in 180 of 189 (95%) of the basins. These results are however difficult to compare directly due to the different products chosen and the different basins considered. In terms of the datasets
considered, we chose to limit our study to remote sensing products, excluding land surface models, station based gridded products as well as reanalysis products. This differs from the two aforementioned studies as our goal is to specifically investigate the remote sensing products and work with independent datasets.

Our study, while it considers the largest number of catchments, was limited to those with GRDC station data available over our time period of interest which excluded some large basins. On the other hand, many smaller catchments were considered,
including nested catchments where multiple stations were available. Areas with more dense gauging networks are therefore overrepresented in our study and these correlate with particular catchment characteristics (for instance climate zone) which can influence the ability of remote sensing to close the water balance as will be seen in Sect 3.3.

## 3.2 Results per product and product combination

For the product combinations based on the GPM rainfall product, an average of 925 time series NSE values could be calculated
per combination, while for the combinations based on the TRMM and CHIRPS products, an average of 599 NSE values per combination could be calculated (due to the smaller spatial coverage of these TRMM and CHIRPS).

The median NSE values for all GRDC stations available for the 45 possible product combinations are presented in the Appendix 1. On average, for all combinations of products and all available GRDC stations, 43.4% of the generated discharge time series achieved a positive NSE value. The best performing combination was CHIRPS – MOD16 – JPL which yielded
53.8% of positive NSE values while the worst, GPM – SEBS – GFZ/JPL, yielded 34.7% of positive NSE values. Only an average of 3.2% of the discharge time series generated reached the threshold of 0.5, with the best combination (CHIRPS - CMRSET – GFZ) reaching this value for 5.9% of stations. The worst performing combination (GPM - GLEAM – GFZ) reached NSE>0.5 for only 1.3% of stations.

In order to make the product combinations more comparable, the same results are presented for 1) all possible time series
(columns A in Appendix 1) and 2) for only those stations for which all products could be used (columns B in Appendix 1). The main consequence of this is that the high latitude stations which are only covered by GPM are removed from the analysis. When selecting only stations covered by all products, the combination with the highest number of positive NSE values was GPM - CMRSET - JPL with 56% of stations reaching positive median NSE values.

Table 3 shows that the NSE of the computed discharge is most sensitive to the choice of ET product. With median NSE values
ranging from -0.07 to 0.01. The ET product with the highest median NSE and number of NSE series with values above 0 is MOD16. The product with the highest number of series producing NSE values above 0.5 is SSEBOP (followed closely by CMRSET). Precipitation has the second largest impact on NSE, with median values between -.01 and -.02. GPM has the




highest number of series with NSE values above 0, while CHIRPS produce the highest number of series with NSE values

above .5. The computed NSE was not found to be sensitive to the choice of GRACE solution used.

**Table 3: Median NSE for time series containing specific products as well as number of time series with positive (n. NSE>0), NSE above 0.5 (n. NSE>0.5) and total number of time series using the product (n. series). Series have been limited to those covered by all product combinations (591 GRDC stations).**

| Variable | Product | Median NSE | n. NSE> 0 | | n. NSE > .5 | | n. series |
|---|---|---|---|---|---|---|---|
| P | TRMM | -0.01 | 4177 | (46%) | 247 | (3%) | 9000 |
| | GPM | -0.01 | 4291 | (31%) | 307 | (2%) | 13872 |
| | CHIRPS | -0.02 | 3997 | (45%) | 392 | (4%) | 8961 |
| ET | SSEBOP | 0 | 2574 | (40%) | 259 | (4%) | 6399 |
| | MOD16 | 0.01 | 2798 | (44%) | 170 | (3%) | 6321 |
| | SEBS | -0.07 | 2219 | (35%) | 158 | (2%) | 6342 |
| | GLEAM | -0.02 | 2281 | (36%) | 107 | (2%) | 6381 |
| | CMRSET | -0.01 | 2593 | (41%) | 252 | (4%) | 6390 |
| GRACE | JPL | -0.01 | 4179 | (39%) | 311 | (3%) | 10611 |
| | CSR | -0.01 | 4142 | (39%) | 316 | (3%) | 10611 |
| | GFZ | -0.01 | 4144 | (39%) | 319 | (3%) | 10611 |

The precipitation and ET products used in the best performing combination for each station are shown in Figure 4 and Figure

5. Because of the low sensitivity of NSE to storage change solution, no map was generated for the different storage change

products.

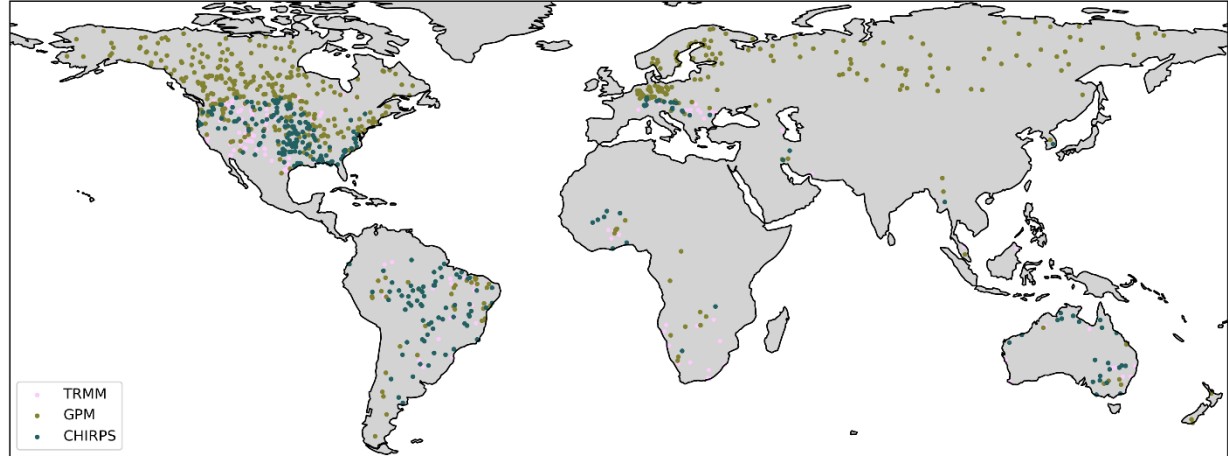

**Figure 4: Precipitation product used in combination with highest NSE at station. Note that GPM is the only product available for latitudes >50°.**





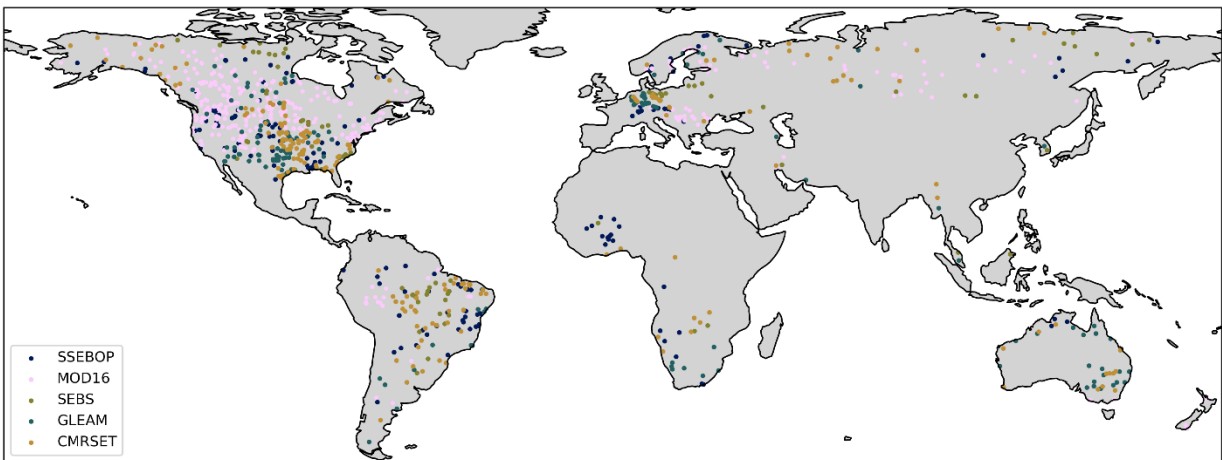

**Figure 5: ET product used in combination with highest NSE at station**

These results show that no single product or combination consistently outperformed others when it comes to the closure of the water balance. This is consistent with findings of previous studies (Lehmann et al., 2022; Lorenz et al., 2014). Some geographic patterns in the better performing products appear in Figure 4 and Figure 5 and will be discussed in the context of the catchment characteristics in the following section.

## 3.3 Results per catchment characteristic

For each of the continuous catchment characteristics listed in Table 2, correlations between the characteristic and the NSE at the GRDC station were computed. Figure 6 shows a summary of the correlations found for all product combinations and the catchment characteristics.

Presence or absence of correlation as well as whether the correlation is positive or negative tends to be consistent across product combinations, except for some combinations containing ET from MODIS, specifically when combined with CHIRPS or TRMM.



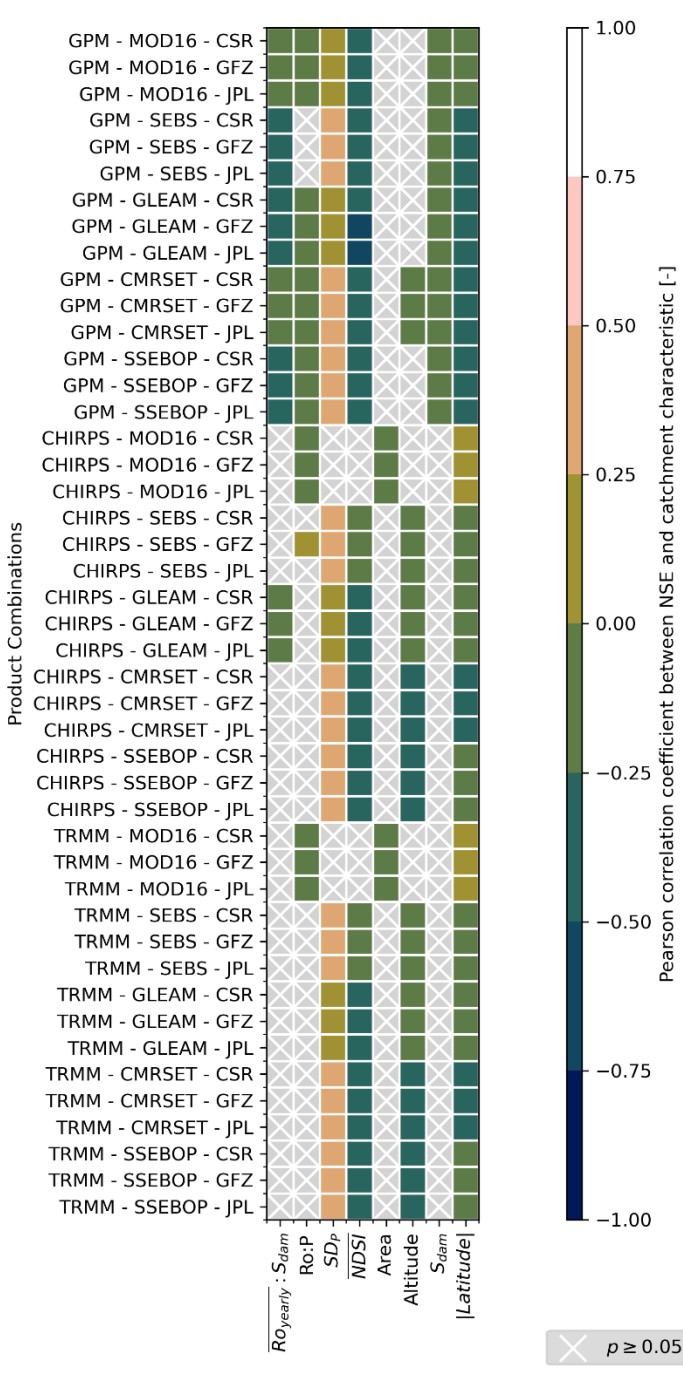

**Figure 6: Pearson correlations for different product combinations between the NSEs of catchments and characteristics of those catchments. See Table 2 for an overview of the catchment characteristics**





Of the catchment characteristics described by a continuous variable, seasonality shows the strongest correlation with the NSE of the discharge. Out of the 45 product combinations, 39 showed a significant correlation with the standard deviation of precipitation. The 6 combinations not showing a correlation all use MODIS as the ET product. It should be noted that
precipitation from GPM was used to compute seasonality, meaning that errors and uncertainties in GPM data could affect catchment classification. The influence of seasonality is in agreement with the findings of Lorenz et al. (2014) who found that the closure of the water balance can be better achieved in basins with a strong seasonal precipitation signal. Lorenz et al. (2014) observed that in catchments with low seasonal runoff variability, the biases in the different input datasets prevented the accurate computation of runoff.

Snow cover has the strongest negative correlation with NSE. NDSI shows a significant correlation for 39 of the 45 product combinations. Combinations including MODIS ET and TRMM or CHIRPS precipitation are the only ones for which no correlation was found. Altitude at the gauging station, which is correlated to snow cover for smaller basins, shows a weaker negative correlation with NSE. The strong negative correlation with snow could be due to multiple factors. For instance, snow retrievals have lower accuracies as compared to liquid precipitation retrievals from satellite and precipitation retrievals are less
accurate over frozen ground (Tang et al., 2020; Tian et al., 2014; Tian and Peters-Lidard, 2010), ET products may not capture the process of sublimation as well as other types of ET (see e.g. Xu et al., 2019), and the snow storage variations which drive discharge timing in some catchments may not be adequately captured by GRACE. Analysis of runoff versus discharge totals over hydrological years, rather than monthly could mitigate the snow storage issue. A similar analysis with more recent data should also be carried out to check if better results for catchments further from the equator (>50°N and >50°S) can be obtained,
as the GPM data from the TRMM era (pre-2014) for higher latitudes is considered partial coverage. The GPM core observatory also has higher sensitivity to snowfall than earlier sensors (Behrangi et al., 2018) and was only launched in 2014.

Latitude also shows a correlation with NSE for all product combinations. This correlation is negative for the same 39 product combinations as snow cover, with the remaining 6 combinations showing a positive correlation. GPM shows a stronger negative correlation with latitude and snow cover than the other products in Figure 6. This effect however disappears when
limiting the analysis to catchments located between 50°S and 50°N where the other precipitation products are available, with the average Pearson correlation coefficient going from -0.28 to -0.16 (the average values for the TRMM and CHIRPS combinations are -0.16 and -0.12 respectively). This negative correlation was expected based on the more extensive snow cover and frozen ground found further from the equator which negatively impacts performance for both P and ET products as explained above.

Dam storage capacity shows a negative correlation with NSE only for product combinations using GPM as a precipitation product. This is also the case for dam storage relative to total runoff. For other combinations, no significant correlations were found.

Runoff ratio shows a negative correlation with NSE for 18 of the 45 combinations, and a positive correlation for only 1 of the 45. Runoff ratio is computed as the ratio of discharge from GRDC and precipitation from GPM, and the maximum value found
was 42, indicating potentially erroneous data or a strong proportion of discharge originating from storage depletion or inter-





basin transfers. Inter-basin transfers in particular would not be represented in our computation of runoff. The runoff ratio was found to be above 1 for 103 stations (out of 937). When excluding those stations, the correlation becomes positive for 15 of the combinations and negative for 27.

No strong correlation was found between drainage area and NSE of the RS-derived runoff. Only 6 of the 45 product combinations, all using MOD16 as ET, show a correlation which is negative. This was unexpected as the GRACE data in particular is expected to perform better for larger catchments. The lack of correlation between NSE and catchment area is surprising as the storage change component from GRACE is expected to perform better over larger catchments, particularly because we limited the catchment size here to catchments larger than 10,000 km² while GRACE is expected to produce reliable estimates of storage change for catchments with areas of more than 50,000 km² (Biancamaria et al., 2019). Sahoo et al. (2011) and Lehmann et al. (2022) similarly found no correlation between basin area and water balance closure though their studies were limited to 10 very large basins and basins with areas larger than 65 000 km² respectively.

Results for the two discrete variables (dominant land cover type and dominant climate class) are shown in Table 4, Table 5, Table 6 and Table 7.

Variability was found between the results for different land cover types. Results for basins with dominant LU codes 40 and 50 (both types of broadleaved forests, see Table 4) perform better than other land cover types, they are the only categories for which median NSE is positive.

Some land cover classes, for example *Open (15-40%) needleleaved deciduous or evergreen forest (>5m)* (class 90), perform particularly poorly, which can be expected as these have a near complete overlap with higher latitude areas. MOD16 performs better than other products in this LC class with a median NSE value of -0.11 while combinations using the other ET products produces median NSE values between -0.59 and -1.55 (Table 5).

Variability is also observed between climate zones, with tropical (median NSE=.11, and median NSE for tropical monsoon .28, see Table 6 and Appendix 1 for the detailed results per climate zone) and temperate zones (median NSE=.06) performing better than arid (median NSE=-.09) and continental zones (median NSE= -.08). The SSEBop and CMRSET products produce the highest NSE values in tropical climates, with median NSE values of 0.17, followed by SEBS at 0.13 (Table 7). In temperate zones, using GPM produces the highest median NSE values of 0.11. Lehmann et al. (2022) also analyzed the water balance closure by climate zone and found that errors were relatively consistent within zones with some exceptions. As in this study, the best performance was observed in the "equatorial rain forest/monsoon" zone. This result is also in agreement with the influence of seasonality of rainfall discussed above and observed by Lorenz et al. (2014). Sahoo et al. (2011) on the other hand did not find consistent behavior based on climate zone.





**Table 4: NSE values for basins classified by dominant land cover class (LCC) as number of time series with positive (n. NSE>0), NSE above 0.5 (n. NSE>0.5), total number of time series with the corresponding land cover (n. series) and corresponding number of catchments (n. catchments)**

| LCC | Land Cover description GlobCover | Median NSE | n. NSE >0 | | n. NSE >0.5 | | n. series | n. catchments |
|---|---|---|---|---|---|---|---|---|
| 14 | Rainfed croplands | -0.02 | 867 | (45%) | 14 | (1%) | 1920 | 65 |
| 20 | Mosaic cropland (50-70%) / vegetation (grassland/shrubland/forest) (20-50%) | -0.07 | 445 | (41%) | 13 | (1%) | 1080 | 33 |
| 30 | Mosaic vegetation (grassland/shrubland/forest) (50-70%) / cropland (20-50%) | 0.00 | 1088 | (49%) | 4 | (0%) | 2220 | 56 |
| 40 | Closed to open (>15%) broadleaved evergreen or semi-deciduous forest (>5m) | 0.19 | 2602 | (72%) | 619 | (17%) | 3603 | 83 |
| 50 | Closed (>40%) broadleaved deciduous forest (>5m) | 0.16 | 4347 | (72%) | 218 | (4%) | 6054 | 162 |
| 60 | Open (15-40%) broadleaved deciduous forest/woodland (>5m) | -0.09 | 165 | (40%) | 0 | (0%) | 417 | 19 |
| 70 | Closed (>40%) needleleaved evergreen forest (>5m) | -0.17 | 658 | (25%) | 12 | (0%) | 2631 | 62 |
| 90 | Open (15-40%) needleleaved deciduous or evergreen forest (>5m) | -0.85 | 358 | (14%) | 20 | (1%) | 2619 | 173 |
| 100 | Closed to open (>15%) mixed broadleaved and needleleaved forest (>5m) | -0.70 | 23 | (6%) | 0 | (0%) | 393 | 17 |
| 110 | Mosaic forest or shrubland (50-70%) / grassland (20-50%) | -2.91 | 0 | (0%) | 0 | (0%) | 30 | 2 |
| 120 | Mosaic grassland (50-70%) / forest or shrubland (20-50%) | -0.27 | 51 | (17%) | 0 | (0%) | 300 | 8 |
| 130 | Closed to open (>15%) (broadleaved or needleleaved, evergreen or deciduous) shrubland (<5m) | -0.05 | 1364 | (33%) | 66 | (2%) | 4080 | 95 |
| 140 | Closed to open (>15%) herbaceous vegetation (grassland, savannas or lichens/mosses) | -0.05 | 1263 | (28%) | 0 | (0%) | 4554 | 116 |
| 150 | Sparse (<15%) vegetation | -0.75 | 312 | (20%) | 0 | (0%) | 1542 | 75 |
| 180 | Closed to open (>15%) grassland or woody vegetation on regularly flooded or waterlogged soil - Fresh, brackish or saline water | -0.07 | 6 | (17%) | 0 | (0%) | 36 | 1 |
| 200 | Bare areas | -0.35 | 21 | (7%) | 0 | (0%) | 288 | 7 |
| 210 | Water bodies | -0.50 | 0 | (0%) | 0 | (0%) | 66 | 4 |



**Table 5: Median NSE values per product and per dominant LU class. Cells in italic bold have median values>0, and cells in bold >0.1. Empty cells represent a category where a specific product is not available.**

| Dominant Land Cover Class | TRMM | GPM | CHIRPS | SSEBOP | MOD16 | SEBS | GLEAM | CMRSET | JPL | CSR | GFZ |
|---|---|---|---|---|---|---|---|---|---|---|---|
| | **Med. NSE** | **Med. NSE** | **Med. NSE** | **Med. NSE** | **Med. NSE** | **Med. NSE** | **Med. NSE** | **Med. NSE** | **Med. NSE** | **Med. NSE** | **Med. NSE** |
| **14** | 0.00 | -0.01 | -0.03 | -0.14 | -0.03 | -0.02 | 0.00 | ***0.01*** | -0.02 | -0.01 | -0.02 |
| **20** | -0.01 | -0.12 | -0.07 | ***0.04*** | -0.32 | -0.03 | -0.11 | -0.06 | -0.07 | -0.06 | -0.07 |
| **30** | -0.01 | 0.00 | 0.00 | -0.01 | 0.00 | -0.28 | ***0.03*** | ***0.02*** | 0.00 | 0.00 | 0.00 |
| **40** | **0.20** | **0.18** | **0.20** | **0.21** | **0.21** | **0.21** | ***0.07*** | **0.27** | **0.19** | **0.19** | **0.20** |
| **50** | **0.16** | **0.18** | **0.13** | **0.19** | **0.19** | **0.19** | ***0.05*** | ***0.10*** | **0.16** | **0.16** | **0.15** |
| **60** | -0.07 | -0.13 | -0.07 | ***0.07*** | -0.46 | -0.04 | -0.13 | ***0.01*** | -0.10 | -0.09 | -0.09 |
| **70** | -0.19 | -0.15 | -0.16 | -0.12 | ***0.02*** | -0.16 | -0.17 | -1.43 | -0.16 | -0.17 | -0.16 |
| **90** | -0.35 | -0.87 | -0.40 | -0.59 | -0.11 | -1.55 | -1.12 | -0.96 | -0.88 | -0.83 | -0.85 |
| **100** | -0.77 | -0.55 | -0.98 | -0.41 | -0.21 | -0.81 | -0.79 | -1.10 | -0.70 | -0.71 | -0.68 |
| **110** | | -2.91 | | -3.72 | -7.15 | -5.56 | -4.11 | -2.41 | -2.90 | -2.99 | -3.08 |
| **120** | -0.14 | -0.35 | -0.12 | -0.29 | -0.50 | -1.10 | -0.04 | -0.03 | -0.28 | -0.28 | -0.23 |
| **130** | -0.03 | -0.06 | -0.04 | -0.03 | -0.01 | -0.32 | -0.01 | -0.09 | -0.05 | -0.05 | -0.05 |
| **140** | -0.04 | -0.06 | -0.04 | -0.04 | -0.05 | -0.35 | -0.02 | -0.02 | -0.05 | -0.05 | -0.05 |
| **150** | -0.20 | -0.98 | -0.33 | -0.84 | -0.27 | -0.79 | -1.32 | -0.79 | -0.75 | -0.77 | -0.74 |
| **180** | -0.09 | -0.08 | ***0.00*** | -0.02 | -0.66 | | -0.10 | -0.06 | -0.06 | -0.07 | -0.07 |
| **200** | -0.35 | -0.41 | -0.32 | -0.06 | -0.27 | -0.33 | -0.33 | -2.01 | -0.36 | -0.34 | -0.34 |
| **210** | -0.67 | -0.50 | -0.80 | -0.22 | -0.26 | -0.61 | -0.70 | -1.24 | -0.52 | -0.51 | -0.49 |

**Table 6: NSE values for basins classified by climate class**

| **Climate class** | | **Median NSE** | **n. NSE >0** | | **n. NSE >0.5** | | **n. series** | **n. catchments** |
|---|---|---|---|---|---|---|---|---|
| **A** | **Tropical** | 0.11 | 3392 | (64%) | 584 | (11%) | 5283 | 127 |
| **B** | **Arid** | -0.09 | 1464 | (23%) | 53 | (1%) | 6474 | 153 |
| **C** | **Temperate** | 0.06 | 3623 | (60%) | 181 | (3%) | 6054 | 162 |
| **D** | **Continental** | -0.08 | 4883 | (36%) | 148 | (1%) | 13620 | 526 |
| **E** | **Polar** | 0.02 | 208 | (52%) | 0 | (0%) | 402 | 11 |

**Table 7: Median NSE values per product and dominant climate class. Cells in italic bold have median values>0, and cells in bold >0.1.**





| | | TRMM | GPM | CHIRPS | SSEBOP | MOD16 | SEBS | GLEAM | CMRSET | JPL | CSR | GFZ |
|---|---|---|---|---|---|---|---|---|---|---|---|---|
| | | Med. NSE | Med. NSE | Med. NSE | Med. NSE | Med. NSE | Med. NSE | Med. NSE | Med. NSE | Med. NSE | Med. NSE | Med. NSE |
| A | Tropical | **0.11** | **0.10** | *0.10* | **0.17** | *0.03* | **0.13** | *0.02* | **0.17** | *0.10* | *0.11* | *0.11* |
| B | Arid | -0.07 | -0.09 | -0.09 | -0.05 | -0.07 | -0.43 | -0.03 | -0.12 | -0.09 | -0.09 | -0.09 |
| C | Temperate | *0.04* | **0.11** | *0.05* | *0.07* | *0.08* | *0.05* | *0.06* | *0.07* | *0.06* | *0.06* | *0.06* |
| D | Continental | -0.02 | -0.18 | -0.04 | -0.09 | *0.02* | -0.15 | -0.12 | -0.19 | -0.08 | -0.08 | -0.09 |
| E | Polar | **0.16** | 0.00 | -0.03 | **0.34** | **0.13** | **0.35** | -0.32 | -0.28 | *0.01* | *0.01* | *0.02* |

## 4 Conclusions and perspectives

In this study, we analyzed the closure of the water balance at the monthly time-scale for catchments of more than $100\,000\,\text{km}^2$ by using remote sensing to compute runoff and comparing the computed runoff to in-situ measurements of discharge from the GRDC using the Nash-Sutcliffe Efficiency as the performance metric. We computed the results for 45 different remote sensing product combinations at 591 to 937 gauging stations and analyzed the results through the lens of both the remote sensing products and of 11 catchment characteristics which we computed globally.

Overall, a positive NSE could be reached for at least one product combination for 72.5% of the stations considered. While some product combinations showed better results than others, no one combination or product stood out as systematically performing better than the others. Correlations were found between the NSE values obtained and the ability of remote sensing to close the water balance between areas with different precipitation patterns, in areas with large snow-cover, in different climatic zones and in areas with different dominant land cover classes. This highlights the importance of validating RS products

widely. In particular, our results point to the necessity of the improvement of products in continental and arid climate zones and some land covers.

While a number of catchments characteristics were analyzed, these are not exhaustive and for those chosen could have also been computed differently. For example, for larger basins, selecting only one land use category as representative can obscure some differences, and using percentages of area under different types of vegetation may help to further refine results. The same

may be considered for climate class. Some additional characteristics which could be interesting to investigate are percentage of area under irrigation in particular for potentially differentiating the different ET products and as a measure of the degree of alteration. One limitation for such an analysis would be the accuracy of global irrigation maps.

Many satellite products are also calibrated in specific areas though it is not always straightforward to obtain this information consistently. It would be very interesting to assess how different the performance is in areas where calibration activities are

carried out versus others and how this impacts the choice of product. These areas could also be correlated with areas with a high density of GRDC stations. Efforts to collect discharge data in underrepresented areas should be undertaken to be included in future studies.



**Author contribution**

The study was designed by Claire Michailovsky and Bert Coerver. The code to process and analyze data was developed and
run by Bert Coerver. The article was written by Claire Michailovsky and Bert Coerver with input from all the co-authors.

**Competing interests**

The author Graham Jewitt is a member of the editorial board of HESS. The peer-review process was guided by an independent
editor, and the authors have no other competing interests to declare.

**Acknowledgements**

This research was supported by the Water and Development Partnership Programme (DUPC2) of IHE Delft Institute for Water
Education, funded by the Dutch ministry of Foreign Affairs.

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





## Appendix 1: Full result tables for all combinations and by climate zone

**Table A1 - Median NSE values for the 45 product combinations. n. NSE>x is the number of time series for which NSE>x and n. catchments is the number of series considered for the specific combination (1 per catchment). The results are presented both for all GRDC stations available for each combination (A.) and for the GRDC stations common to all product combinations (B.)**

| Product Combination | median NSE | n. NSE>0.5 | n. NSE>0 | n. catchmts | median NSE | n. NSE>0.5 | n. NSE>0 | n. catchmts |
|---|---|---|---|---|---|---|---|---|
| | | A: For all possible series | | | | B: For series common to all products | | |
| TRMM - SSEBOP - JPL | 0.00 | 24 | 295 | 601 | 0.00 | 24 | 295 | 591 |
| TRMM - SSEBOP - GFZ | -0.01 | 24 | 285 | 601 | 0.00 | 24 | 285 | 591 |
| TRMM - SSEBOP - CSR | 0.00 | 25 | 292 | 601 | 0.00 | 25 | 292 | 591 |
| TRMM - CMRSET - JPL | -0.01 | 21 | 291 | 604 | -0.01 | 21 | 288 | 591 |
| TRMM - CMRSET - GFZ | -0.01 | 22 | 288 | 604 | -0.01 | 22 | 285 | 591 |
| TRMM - CMRSET - CSR | -0.01 | 21 | 291 | 604 | 0.00 | 21 | 288 | 591 |
| TRMM - GLEAM - JPL | -0.01 | 10 | 257 | 599 | -0.01 | 10 | 255 | 591 |
| TRMM - GLEAM - GFZ | -0.02 | 10 | 253 | 599 | -0.02 | 10 | 251 | 591 |
| TRMM - GLEAM - CSR | -0.02 | 10 | 256 | 599 | -0.01 | 10 | 253 | 591 |
| TRMM - SEBS - JPL | -0.05 | 12 | 256 | 601 | -0.05 | 12 | 254 | 591 |
| TRMM - SEBS - GFZ | -0.06 | 14 | 253 | 601 | -0.06 | 14 | 251 | 591 |
| TRMM - SEBS - CSR | -0.06 | 12 | 256 | 601 | -0.06 | 12 | 252 | 591 |
| TRMM - MOD16 - JPL | 0.01 | 14 | 309 | 595 | 0.01 | 14 | 309 | 591 |
| TRMM - MOD16 - GFZ | 0.01 | 14 | 311 | 595 | 0.01 | 14 | 311 | 591 |
| TRMM - MOD16 - CSR | 0.01 | 14 | 308 | 595 | 0.01 | 14 | 308 | 591 |
| CHIRPS - SSEBOP - JPL | -0.01 | 35 | 288 | 601 | -0.01 | 35 | 286 | 591 |
| CHIRPS - SSEBOP - GFZ | 0.00 | 35 | 289 | 601 | 0.00 | 35 | 287 | 591 |
| CHIRPS - SSEBOP - CSR | -0.01 | 35 | 289 | 601 | 0.00 | 35 | 287 | 591 |
| CHIRPS - CMRSET - JPL | -0.04 | 33 | 249 | 598 | -0.04 | 33 | 248 | 591 |
| CHIRPS - CMRSET - GFZ | -0.05 | 35 | 248 | 598 | -0.04 | 35 | 247 | 591 |
| CHIRPS - CMRSET - CSR | -0.04 | 33 | 249 | 598 | -0.04 | 33 | 248 | 591 |
| CHIRPS - GLEAM - JPL | -0.01 | 11 | 257 | 599 | -0.01 | 11 | 255 | 591 |
| CHIRPS - GLEAM - GFZ | -0.02 | 13 | 252 | 599 | -0.02 | 13 | 250 | 591 |
| CHIRPS - GLEAM - CSR | -0.02 | 12 | 246 | 599 | -0.02 | 12 | 245 | 591 |
| CHIRPS - SEBS - JPL | -0.08 | 24 | 235 | 594 | -0.08 | 24 | 235 | 591 |
| CHIRPS - SEBS - GFZ | -0.09 | 23 | 229 | 594 | -0.09 | 23 | 229 | 591 |
| CHIRPS - SEBS - CSR | -0.09 | 24 | 234 | 594 | -0.09 | 24 | 234 | 591 |
| CHIRPS - MOD16 - JPL | 0.01 | 26 | 320 | 595 | 0.01 | 26 | 319 | 591 |
| CHIRPS - MOD16 - GFZ | 0.01 | 27 | 315 | 595 | 0.01 | 27 | 314 | 591 |
| CHIRPS - MOD16 - CSR | 0.01 | 26 | 314 | 595 | 0.01 | 26 | 313 | 591 |



| | | | | | | | |
|---|---|---|---|---|---|---|---|
| GPM - SSEBOP - JPL | -0.06 | 27 | 339 | 931 | -0.01 | 27 | 284 | 591 |
| GPM - SSEBOP - GFZ | -0.06 | 27 | 339 | 931 | -0.01 | 26 | 282 | 591 |
| GPM - SSEBOP - CSR | -0.06 | 29 | 331 | 931 | -0.01 | 28 | 276 | 591 |
| GPM - CMRSET - JPL | -0.07 | 28 | 379 | 928 | 0.03 | 28 | 331 | 591 |
| GPM - CMRSET - GFZ | -0.07 | 31 | 377 | 928 | 0.02 | 31 | 329 | 591 |
| GPM - CMRSET - CSR | -0.08 | 28 | 378 | 928 | 0.03 | 28 | 329 | 591 |
| GPM - GLEAM - JPL | -0.06 | 14 | 327 | 929 | -0.01 | 14 | 258 | 591 |
| GPM - GLEAM - GFZ | -0.06 | 12 | 325 | 929 | -0.02 | 12 | 258 | 591 |
| GPM - GLEAM - CSR | -0.06 | 15 | 327 | 929 | -0.02 | 15 | 256 | 591 |
| GPM - SEBS - JPL | -0.22 | 16 | 319 | 919 | -0.07 | 16 | 253 | 591 |
| GPM - SEBS - GFZ | -0.23 | 17 | 319 | 919 | -0.07 | 17 | 256 | 591 |
| GPM - SEBS - CSR | -0.22 | 16 | 320 | 919 | -0.07 | 16 | 255 | 591 |
| GPM - MOD16 - JPL | -0.02 | 23 | 425 | 917 | 0.02 | 16 | 309 | 591 |
| GPM - MOD16 - GFZ | -0.02 | 20 | 423 | 917 | 0.01 | 16 | 309 | 591 |
| GPM - MOD16 - CSR | -0.02 | 24 | 427 | 917 | 0.01 | 17 | 306 | 591 |





**Table A2: Full results by climate zone. NSE>x is the number of time series for which NSE>x and n.series the number of time-series produced for each climate class and n.catchments is the number of catchments located in the different climate classes.**

| | Climate class | | Median NSE | n.NSE>0 | n.NSE>0.5 | n. series | n. catchments |
|---|---|---|---|---|---|---|---|
| 1 | Af | Tropical rainforest climate | 0.15 | 300 | 52 | 450 | 10 |
| 2 | Am | Tropical monsoon climate | 0.28 | 642 | 284 | 936 | 21 |
| 3 | Aw/As | Tropical wet and dry or savanna | 0.08 | 2450 | 248 | 3897 | 96 |
| 4 | BWh | Hot desert climate | -0.11 | 126 | 0 | 579 | 14 |
| 5 | BWk | Cold desert climate | -0.20 | 27 | 0 | 315 | 7 |
| 6 | BSh | Hot semi-arid climate | -0.04 | 448 | 53 | 1137 | 28 |
| 7 | BSk | Cold semi-arid climate | -0.09 | 863 | 0 | 4443 | 104 |
| 8 | Csa | Hot-summer Mediterranean climate | -0.04 | 33 | 0 | 90 | 2 |
| 9 | Csb | Warm-summer Mediterranean climate | -0.12 | 166 | 30 | 450 | 10 |
| 11 | Cwa | Monsoon-influenced humid subtropical climate | 0.05 | 218 | 51 | 396 | 21 |
| 12 | Cwb | Monsoon-influenced temperate oceanic climate | -0.06 | 85 | 0 | 225 | 5 |
| 14 | Cfa | Humid subtropical climate | 0.11 | 2347 | 90 | 3591 | 89 |
| 15 | Cfb | Temperate oceanic climate | 0.06 | 774 | 10 | 1302 | 35 |
| 18 | Dsb | Mediterranean-influenced warm-summer humid continental climate | -0.85 | 54 | 0 | 345 | 9 |
| 19 | Dsc | Mediterranean-influenced subarctic climate | -0.09 | 30 | 4 | 135 | 7 |
| 21 | Dwa | Monsoon-influenced hot-summer humid continental climate | 0.74 | 45 | 45 | 45 | 3 |
| 22 | Dwb | Monsoon-influenced warm-summer humid continental climate | -0.04 | 26 | 0 | 105 | 3 |
| 23 | Dwc | Monsoon-influenced subarctic climate | -0.86 | 16 | 0 | 120 | 8 |
| 24 | Dwd | Monsoon-influenced extremely cold subarctic climate | -0.72 | 5 | 0 | 30 | 3 |
| 25 | Dfa | Hot-summer humid continental climate | 0.13 | 1650 | 58 | 2295 | 51 |
| 26 | Dfb | Warm-summer humid continental | -0.07 | 2630 | 29 | 7503 | 248 |
| 27 | Dfc | Subarctic climate | -0.89 | 424 | 12 | 2970 | 189 |
| 28 | Dfd | Extremely cold subarctic climate | -1.51 | 3 | 0 | 72 | 5 |
| 29 | ET | Tundra climate | 0.06 | 172 | 0 | 312 | 9 |
| 31 | EF | Ice cap climate | -0.13 | 36 | 0 | 90 | 2 |