# Peer review of "Investigating sources of variability in closing the terrestrial water balance with remote sensing"

_Hydrology and Earth System Sciences, 2023_

## Author Response (AR1)

**Answers to reviews are in green.**

**Reviewer #1**

Closing water balance is an important issue for the hydrology community. This paper analyzed the water balance for 591 catchments across the globe to investigate correlations between catchment characteristics and the quality of RS based water balance estimates. It demonstrates whether specific products performed better than others in certain conditions. This will provide important referee for other scientists. However, to make their conclusion to be more rigorous, I suggest to include other dataset, which might influence some statements in this paper. The presentation is well structured and the quality of writing is good. I suggest it could be published after this revision.

This study uses SEBS ET data which is produced by the SEBS model updated by Chen et al. 2013. The SEBS ET data used in this study is produced by a monthly input data. This model version has problems over the high canopy which is already reported in Chen et al. 2019. Chen et al. 2019 paper has solved the low estimation of sensible heat flux over the forest area. After the model revision, a daily global ET data at 0.05 degree has been produced by Chen et al. 2021 JGR. The daily ET data has been shared through:

https://data.tpdc.ac.cn/en/data/df4005fb-9449-4760-8e8a-09727df9fe36/

This ET is a seamless data based on energy balance structure of SEBS. I suggested this dataset should be included in this analysis, since it has a high potential to change the conclusions on the dataset combination performance. In addition, the SEBS monthly ET has missing pixels over the Sahara, Arabian desert, Taiga in Canada and Rsussia. Meanwhile the updated daily ET data in Chen et al. 2021 is a gap-free daily ET data. Both ET dataset has the same spatial resolution. Hereby, I believe that this dataset will benefit this study.

Chen, X., Massman, W.J. and Su, B.Z., 2019. A column canopy-air turbulent diffusion method for different canopy structures. Journal of Geophysical Research: Atmospheres, 124: 488–506.

Chen, X., Su, Z., Ma, Y., Trigo, I. and Gentine, P., 2021. Remote Sensing of Global Daily Evapotranspiration based on a Surface Energy Balance Method and Reanalysis Data. Journal of Geophysical Research: Atmospheres, 126(16): e2020JD032873.

**RESPONSE:**

Thank you for your comments.

While we do not expect to provide a definitive ranking of products under different conditions (new products and versions are frequently released and such a ranking would rapidly be obsolete), we have collected the newer SEBS dataset and included it in our analysis. All tables and figures have been changed to include the newer dataset.

**Reviewer #2**

This work by Michailovsky et al. was motivated by the variability of the degree of closure of the water balance for different catchment characteristics and for the use of different RS products in previous studies. Their goal was twofold: (1) testing which combinations of RS products (precipitation, ET and water storage change products) best reproduce in situ measurements of

discharge for multiple catchments, and (2) identifying catchment characteristics which can explain the achieved degree of closure of the water balance. They used 45 combinations of RS products, and used the water balance to compare the resulting monthly discharge against in situ measurements of 591 catchments. Finally, they used 11 quantifiable catchment characteristics to evaluate how well these monthly discharges match.

The article is of great interest to all those who (aim to) use remote sensing products for closing the water balance in order to estimate discharge, particularly in poorly gauged river basins. The article is well-structured and well-written. Given the relevance of this work, I recommend to publish this article after considering the comments below.

Thank you for your comments and recommendation.

General comments

C1.I believe the authors could bring their analysis one step further by better discussing why some RS product combinations perform better than others for specific areas / catchment characteristics.

In order to understand why certain products perform better in certain catchments, one needs to know what are the general differences between the products for the same variable (i.e. measurement principle, data sources, idea of the algorithm, etc.). A brief explanation of the differences between each product could be added to paragraphs 2.1.1, 2.1.2 and 2.1.3. In chapter 3, the authors could refer to these differences when explaining why for example one ET product performs better than another ET product, in combination with the same P and dS/dT products, in certain types of catchments. This would be very valuable information for those who have to choose products to estimate discharge.

We have added the methods employed by the different ET models to section 2.1.2.

We agree that differences between products could give useful insight into the different model performances but the details go beyond the scope of our study and would need to integrate equations, inputs and parametrizations of different models and would be an interesting follow up to this paper.

C2. Consider adding a Figure/Table which shows the best performing product combinations for certain common combinations of catchment characteristics. This would be most informative for water managers of ungauged basins.

Some of the products we used in the study are no longer produced in the versions used and there are continuous improvements to the algorithms used to produce the data. We therefore do not aim to produce a final classification of best/worst products which would likely be obsolete by the time of publication, but rather provide a consideration of the factors which influence the closure of the water balance. We believe that this is more valuable to users than a ranking.

C3. Most of the data sets used are products from missions that are not in orbit any more. I think it is relevant for the target users to describe what are the current / future missions and products, which are comparable to the missions used for the current analysis. Some of this information is given in a footnote in Table 1 (GRACE-FO, TMPA), but given its relevance for the users of this work, I think this deserves a paragraph in the text.

For most of the products newer versions are simply available through the same providers (e.g. newer versions of GPM, GRACE or SSEBop). As products and versions evolve quickly, we do not aim to give an overview of all current products as such lists can quickly become obsolete.C4. Did you consider adding dominant soil type and/or subsurface bedrock depth to the catchment characteristics? These are important factors affecting the discharge of a catchment, particularly when looking at different continents.

We did not add these, but agree they could be of interest. We have added these as suggestions (l. 457-459), in particular for the soil type – we are not aware of a globally available subsurface bedrock depth product which could be used in this study.

Line-specific comments

C5. In line 77-103, a clear overview of previous related studies is given. However, I find the novelty of the current study a bit underexposed in lines 104-107. Both aims that are mentioned (i.e. to investigate both the ability of different combinations of RS products to reproduce in situ measurements of discharge, and to identify catchment characteristics that affect how well the closure of the water balance can be achieved) have been studies before. Here, it is not yet clear what the current study adds to the previous studies, and why this study is necessary. I recommend to formulate the gap in knowledge and added value of the current study more clearly in this section.

We have added to the description of the novelty in lines 108-109.

C6. L54 / L70: Equation 1 and 2 are exactly the same, while Eq. 2 should be a rewritten version of Eq.1.

Yes, thank you for spotting this error. Eq. 2 should be Q = P – ETa – dS/dt and has been changed.

C7. L66 "This is equivalent to the assumption that subsurface fluxes in and out of the basin are negligible". Consider referring to Bouaziz et al. (2018), who disprove this assumption.

Thank you for the reference, we have added a comment on the assumption in the introduction (l. 68-69) and in discussing correlation with catchment area (l. 404-406).

C8. L440: "100,000 km2". Based on previous text, I assume this should be 10,000 km2.

Yes, thank you for spotting this error -it has been corrected.

C9. L233: "We selected 11 RS derived catchment characteristics… . These are summarized in Table 2 …" Table 2 only shows 10 RS catchment characteristics.

Indeed, we have fixed this error throughout the paper.

C10: L395: "This was unexpected as the GRACE data in particular is expected to perform better for larger catchments" Given the next sentence, I think this sentence is redundant.

We agree, we have removed the sentence.

C11. Fig. 5: The colors of SSEBOP and and GLEAM are too similar. It is difficult to distinguish the two. Please adjust the colors.

We have changed the colormap and removed the grey background for improved legibility.

C12. Fig. 6: Consider revising the color scheme used. Make sure the colors for a positive correlation can be clearly distinguished from the colors for a negative correlation. Now, the green color

representing a Pearson correlation of 0-0.25 is similar to the green/blue colors representing a negative correlation.

We had chosen a color scheme that could work in black and white but agree a diverging color scheme adds to legibility and we have adjusted the color scheme.

References

Bouaziz, L., Weerts, A., Schellekens, J., Sprokkereef, E., Stam, J., Savenije, H., & Hrachowitz, M. (2018). Redressing the balance: quantifying net intercatchment groundwater flows. Hydrology and Earth System Sciences, 22(12), 6415-6434.
* * *
**Reviewer #3**

Recommendation:

accept after (some) major revisions

General remarks:

The authors have investigated river discharge derived from various combinations of GRACE gridded products, in combination with precipitation and evapotranspiration products from several sources. The estimated discharge estimates are rigorously compared in terms of their Nash-Sutclife efficiency (NSE) with in situ data from the global river discharge centre. Furthermore, correlations between the resulting NSE and a list of catchment characteristics are compared to see if certain characteristics are linked to better performing discharge estimates.

The authors find an overall median NSE close to zero while less than half of the combinations reach a positive NSE. In the most optimal case, when selecting the best performing combination for each catchment, 72.5 percent of the comparison exhibit a positive NSE. The analysis shows that a one-size-fits-all combination of GRACE and P, ET products can not be found.

I've found the paper interesting to read and I appreciate the rigorous approach to testing all possible data combinations. From my viewpoint, the paper is definitely suitable for publications after consideration of some issues (see below). I assume these are relatively easy to address, but since some may involve redoing some computations I still opted for a major revision.

Thank you very much for your comments and recommendation.

Main issues:

* Relevant context on the GRACE spatial resolution and signal leakage is missing for the particular products used. In the study, gridded GRACE products have been used. This is in principle ok, but it the inherent spatial resolution of the GRACE derived total water storage is important and should be mentioned. The truncation (spherical harmonic) degree of the input gravity field solutions in combination with the strength of the applied spatial filtering will result in highly smoothed fields and signal attenuation (and contamination from nearby sources). I suspect that the smoothing may also

contribute to the lacking correlation found between basins area and the obtainable NSE: below the typical GRACE resolution the results for the sub-catchments will result in essentially the same GRACE times series.

I see 3 ways out of this conundrum: (1) filter the P-ET data with a spatial filter which has similar smoothing as that was is applied GRACE, (2) try to restore the signal in GRACE (see e.g. Vishawakarma et al. 2016 below or similar references), or (3) leave it as is but write a disclaimer in the discussion on how this can effect the results.

Way (2) (then (1)) would be the best but it would essentially constitute a lot of work. I would find (3) also acceptable in light of the, likely larger, errors introduced through the P and ET estimates.

The inherent GRACE resolution was added in the result section l. 402. We do note the issue of signal location and leakage, though not in detail, when discussing GRACE's ability to locate punctual storage changes from dams (l.257) and have added a reference on leakage and catchment size (l. 404-405)

Regarding the suggested options, our goal in this paper is to apply the simplest approach to solving the water balance – following the approach that a standard application user of these products would – and because of this we will be sticking to the use of the gridded GRACE data as is. However, we will make the issues brought up by the reviewer more explicit in the paper.

* How are biases treated in R_estimated and R_observed? If biases are large, some of them may strongly influence the NSE, in particular for arid catchments (bias is larger compared to the standard deviation). Maybe a NSE variant can be computed where the biases are removed?

In the paper, we aim to assess the performance of the products as is, including potential biases. We therefore did not perform any bias corrections on inputs or outputs. We also chose NSE as we wanted the results to be understandable to practitioners and NSE is the most commonly used indicator for hydrological model performance. For this reason, we have kept NSE as the indicator.

Note that at higher latitudes, errors in the Glacial isostatic adjustment trend-correction for GRACE may also manifest themselves as biases in the storage changes, although I suspect that this is not a big issue as the GIA models generally perform well over the Northern Hemisphere (where most of the considered catchments are)

Thank you for this note, we have added a reference to this issue on l.385-389.

* North American nested-catchments may be disproportionally present in the catchment characteristics comparison. The authors mention that several catchments are nested-catchments of their larger parent groups. Is this somehow compensated when computing the correlations of the catchment characteristics? If not I wonder whether e.g. the Missisippi basin, which is well gauged may be overrepresented in the statistics. The authors may consider limiting the amount of subcatchments which are fed into the statistics, if this is an issue.

It is indeed likely that this has an influence on the general statistics – we include info on this in lines 305-308, as well as the fact that these catchments will typically be in (for example) similar climate zones. We chose to use all available data throughout the paper, otherwise questions of which catchments to keep and remove becomes intractable.

* Suggestion: Use histograms (cathment nr. on y versus NSE on x) to visualize and thus understand the distribution of the obtained NSE's. It would also help justify the computation of the correlation

of the NSE (essentially a correlation of a correlation), which implicitly assumes that the computed NSE's are normally distributed.

Thank you for this suggestion, we have included a histogram for better visualization of results (fig. 2 in the text), and you are right, the NSE values are not normally distributed. We therefore decided to compute the Spearman rank correlation as it is non-parametric and does not rely on the assumption of normality. We found that for 6 of the 8 tested continuous catchment characteristics, the general pattern remains similar though with higher correlations and more significant correlations found (see figures below). For the remaining 2 characteristics (Ro_yearly/Sdam and Ro_yearly/P) the pattern is inverted for one, and it is hard to tell for the other (Ro_yearly/P) as Pearson mostly returned non-significant values. We will revise our paper to include the Spearman rather than Pearson results.

[Figure]

Minor issues

* Please mention how aggregate catchment values are obtained from the gridded values. Are grid areas taken constant, or are they latitude weighted?

Thank you for this comment, we had omitted this – all calculations were redone with area-weighting (l. 137-138).

\* I would like to ask the authors to consider releasing their analysis code, to improve reproducibility for others

All code we wrote and used for this study can be found at https://zenodo.org/record/8318720.

References:

Dutt Vishwakarma, B., Devaraju, B., Sneeuw, N., 2016. Minimizing the effects of filtering on catchment scale GRACE solutions. Water Resources Research 52, 5868–5890. https://doi.org/10.1002/2016WR018960
* * *
**Reviewer #4**

This reviewer appreciates the effort of the authors to investigate sources of variability in closing the terrestrial water balance with remote sensing. The novelty of this study is to link the catchment characteristics (and LCC, LU, and climate class) to the performance of different product combinations. Although the manuscript collects multi-source RS water cycle variable datasets, it is unclear why certain product combinations perform better than others. For example, it is unclear the impact of different spatial resolutions of different products on the final results.

This reviewer suggests the authors put more effort into clarifying the above perspective, for example:

Clarifying and quantifying the potential impacts of using RS data from different spatial/temporal resolutions;

In terms of quantifying such impacts, the triple-collocation type of approach could be applied here among multi-source RS data products to understand the relative errors between each other, which will provide further information for identifying the sources of variability in closing the terrestrial water balance using different product combinations.

Thank you for your comments.

In terms of the varying spatio-temporal resolutions of the products we think that as we are running the comparisons on a basin-wide and monthly basis, it would be difficult to identify these issues directly.

Further, as far the triple-collocation method is concerned, one challenge in application is that it requires independent datasets to be used. This is not the case for many of the products as many of them use the same input datasets and/or equations and parameterizations (as noted e.g. on line 153 of the discussion paper for precipitation products).

Minor comments:

Eq.2 looks exactly the same as Eq.1

Thank you for pointing this out, Eq. 2 should read: $Q = P - ETa - dS/dt$. This has been corrected in the text.

Line 36 "the results point to the importance ..." is confusing

Lines 36-37 have been adjusted for clarity.

---

## Author Response (AR2)

Dear Professor Su,

Thank you for handling our paper.

We implemented the technical correction on the SEBS product provided by referee #1 as well as a short paragraph presenting results for the anomalies relative to the mean suggested by referee #3 (paragraph 3.4 and Appendix B).

5 We also updated Fig.3 and Fig.4 to update the color scale and clarify that some values go beyond the lower end of the colorscale.

We include the version of the manuscript with track changes below.

Kind regards,

Claire Michailovsky on behalf of the authors.

[revised manuscript text omitted]